# Facilitation of TRKB Activation by the Angiotensin II Receptor Type-2 (AT2R) Agonist C21

**DOI:** 10.3390/ph14080773

**Published:** 2021-08-06

**Authors:** Liina Laukkanen, Cassiano R. A. F. Diniz, Sebastien Foulquier, Jos Prickaerts, Eero Castrén, Plinio C. Casarotto

**Affiliations:** 1Neuroscience Center, HiLife, University of Helsinki, 00014 Helsinki, Finland; liina.laukkanen@helsinki.fi (L.L.); eero.castren@helsinki.fi (E.C.); 2Department of Pharmacology, Ribeirão Preto School of Medicine, University of São Paulo, Ribeirão Preto 14049-900, SP, Brazil; cassiano.diniz@helsinki.fi; 3Department of Pharmacology and Toxicology, CARIM-School for Cardiovascular Diseases, MHeNS-School for Mental Health and Neuroscience, Maastricht University, 6229 ER Maastricht, The Netherlands; s.foulquier@maastrichtuniversity.nl; 4Department of Psychiatry and Neuropsychology, MHeNS-School for Mental Health and Neuroscience, Maastricht University, 6229 ER Maastricht, The Netherlands; jos.prickaerts@maastrichtuniversity.nl

**Keywords:** compound 21, angiotensin 2 type 2 receptor (AT2R), neurotrophin receptor type 2 (NTRK2), renin-angiotensin system (RAS), fear conditioning

## Abstract

Blockers of angiotensin II type 1 receptor (AT1R) exert antidepressant-like effects by indirectly facilitating the activation of the angiotensin II type 2 receptor (AT2R), which leads to increased surface expression and transactivation of tropomyosin-related kinase B receptors (TRKB). Compound 21 (C21) is a non-peptide AT2R agonist that produces neuroprotective effects. However, the behavioral effects of C21 and its involvement with the brain-derived neurotrophic factor (BDNF)-TRKB system still need further investigation. The aim of the present study was to assess the effect of C21 on the activation of TRKB and its consequences on conditioned fear. The administration of C21 (0.1–10 μM/15 min) increased the surface levels of TRKB but was not sufficient to increase the levels of phosphorylated TRKB (pTRKB) in cultured cortical neurons from rat embryos. Consistent with increased TRKB surface expression, C21 (10 μM/15 min or 3 days) facilitated the effect of BDNF (0.1 ng/mL/15 min) on pTRKB in these cells. In contextual fear conditioning, the freezing time of C21-treated (administered intranasally) wild-type mice was decreased compared to the vehicle-treated group, but no effect of C21 was observed in BDNF.het animals. We observed no effect of C21 in the elevated plus-maze test for anxiety. Taken together, our results indicate that C21 facilitated BDNF effect by increasing the levels of TRKB on the cell surface and reduced the freezing time of mice in a BDNF-dependent manner, but not through a general anxiolytic-like effect.

## 1. Introduction

The renin-angiotensin system (RAS) has been historically implicated in cardiovascular and peripheral fluid homeostasis. However, an increasing number of studies suggest that RAS also plays a role in the central nervous system (CNS). In fact, all RAS components are found within the CNS and function independently of the periphery [1,2,3]. Briefly, the precursor angiotensinogen is cleaved by renin to angiotensin I, which is further converted into angiotensin II (Ang II) by the angiotensin-converting enzyme (ACE).

The majority of ANG2 actions are usually mediated by angiotensin II type 1 and type 2 receptors (AT1R, AT2R). Both receptors are constitutively expressed in brain areas such as the frontal cortex [3,4], which is crucial to the control of the behavioral consequences of stress [5,6]. Accordingly, either acute or chronic stress is sufficient to increase both Ang II and AT1R levels in structures of the hypothalamic-pituitary-adrenal (HPA) axis [7]. Interestingly, inhibitors of the angiotensin-converting enzyme (ACEi) and AT1R antagonists counteract the behavioral consequences of stress exposure [8,9,10], while animals lacking angiotensinogen presented an antidepressant-like phenotype [11].

There is evidence indicating that RAS can modulate the brain levels of both brain-derived neurotrophic factor (BDNF) and its receptor tropomyosin-related kinase B (TRKB). For instance, chronic treatment with telmisartan (AT1 antagonist) prevented retinal damage and a decrease in BDNF levels in an animal model of induced diabetes [12]. Another AT1R antagonist, candesartan, prevented neurological deficits and decreased the infarct brain volume of animals submitted to middle cerebral artery occlusion while increasing TRKB protein and mRNA brain levels [13], and valsartan counteracted the stress consequences on hippocampal and frontal cortex BDNF levels [14]. The facilitation of BDNF/TRKB signaling is crucial for neuroplasticity, including cell differentiation, growth, and synapse formation, and its function has been linked to the mechanism of action of antidepressant drugs [15,16,17,18].

Recently, the antidepressant-like effect of losartan, another AT1R antagonist, has been described to involve TRKB transactivation and increase on the cell surface, an effect likely linked to an indirect AT2R activation after AT1R blockade [19]. However, the evidence of a direct effect from AT2R agonists on behavior and BDNF/TRKB signaling remains limited. Therefore, understanding the functional interaction between AT2R and BDNF/TRKB signaling may facilitate development of new AT2R agonists able to reinstate plasticity in the adult brain, as described for classical antidepressants [17,20,21].

Thus, to extend our previous findings regarding the interaction between AT2R and TRKB [19], we investigated the effects of the AT2R agonist C21, which stimulates neuronal plasticity [22], on the surface exposure and activation of TRKB in vitro, and on the behavioral consequences of exposure to stress.

## 2. Results

### 2.1. C21 Exposes TRKB on the Cell Surface and Facilitates BDNF Effect on TRKB

The administration of C21 increased the levels of TRKB detected on the cell surface [Kruskal-Wallis H(3)= 58.50, *p* < 0.0001]. All doses of C21 (0.1, 1, and 10 μM) were different from the control group (Dunn’s *p* < 0.05; Hedges’ g, 0.1 μM = 1.862; 1 μM = 1.930; 10 μM = 1.418], Figure 1A.

The two-way ANOVA indicated a significant interaction between C21 and BDNF administration at the levels of pTRKB [F(1,33) = 7.144, *p* = 0.0116]. While longer administration of C21 (3 days) increased pTRKB levels per se [treatment: F(1,44) = 423.5, *p* < 0.0001], the facilitation of the BDNF effect was still present [interaction: F(1,44) = 10.22, *p* = 0.0026], Figure 1B,C.

### 2.2. C21 Reduces Behavioral Consequences of Stress

The two-way ANOVA with repeated measures indicated a significant interaction between C21 treatment, genotype and trials even considering the total distance traveled as covariant on freezing time [F(2,58) = 3.631, *p* = 0.033]. Then, we proceeded with two-way ANOVA for each trial individually, as seen in Figure 2, treatment with C21 did not affect the consequences of exposure to shocks in the conditioning session [interaction: F(1,30) = 0.145, *p* = 0.706, Figure 2A]. Two-way ANOVA also indicated a significant interaction between genotype and treatment for the freezing observed in context A (familiar) [F(1,30 = 5.012, *p* = 0.033, Figure 2C] but not in context B (unfamiliar) [F(1,30) = 0.033, *p* = 0.856, Figure 2B]. C21 effectively reduced the time spent in freezing in the familiar context in wt (Fisher’s LSD, *p* < 0.05; Hedges’ g = 2.989) but not in BDNF.het mice (Fisher’s LSD, *p* > 0.05; Hedges’ g = 0.556).

### 2.3. C21 Lacks Anxiolytic-like Effect

As described in Table 1, treatment with C21 did not change any of the parameters assessed in the elevated plus-maze (percent time in the open arms, %OAT: t(10) = 0.914, *p* = 0.383, Hedges’ g = 0.318; percent entries in the open arms, %OAE: t(10) = 1.097, *p* = 0.298, Hedges’ g = 0.227; number of entries in the enclosed arms, EAE: t(10) = −1.351, *p* = 0.206, Hedges’ g = 0.759; total distance traveled: t(10) = 0.032, *p* = 0.975, Hedges’ g = 0.018).

## 3. Discussion

The present study demonstrates that the AT2R agonist C21 is able to increase the surface levels of TRKB in vitro. Interestingly, the treatment of primary neurons with different classes of antidepressants, but not with structurally-related compounds devoid of antidepressant activity, also increases TRKB surface levels [23]. Consistent with our findings, the AT2R antagonist PD123319 decreased the surface levels of TRKB in cultured cortical neurons [19]. In contrast, acute treatment with C21 did not change the levels of pTRKB on cortical cells as observed with another AT2R agonist CGP42112 [19]. The difference between the effect of these AT2R agonists cannot be explained by the small discrepancy of their respective binding affinities: CGP42112 Ki = 0.24 nM [24,25], C21 Ki = 0.4 nM [26,27]. Moreover, even though CGP422112 has 2-fold while C21 only has 20% of Ang II affinity to AT2R, C21 was nevertheless equally effective as Ang II in triggering AT2R-mediated signaling [27]. However, kinetic differences may explain the lack of acute effectiveness of C21 on pTRKB. For instance, AT2R signaling triggered by the C21 administration is slower and shorter in magnitude than by Ang II [28], while CGP 422112 was able to induce comparable effects to Ang II-induced pTRKB [19].

Although not able to activate TRKB per se, we observed that acute administration of C21 enabled the activation of TRKB by an ineffective dose of BDNF in vitro. However, the level of pTRKB was increased when C21 treatment was continued for 3 days. Given the acute effect of C21 on TRKB surface levels, it is plausible that increased levels of TRKB on the cell surface would facilitate the effect of endogenous BDNF along the time, which would result in the increased levels of pTRKB, and putatively increase the interaction with Src kinases such as Fyn as observed previously [19,29,30]. The majority of TRKB resides in vesicles, inaccessible to BDNF, and are transported to the cell surface upon neuronal activity or by stimulation of surface-positioned TRKB [31]. Therefore, it is plausible to expect that after 3 days of C21 treatment active TRKB levels may be increased, as observed here, and boosted by the ineffective BDNF dose. Further, previous data indicated that AT2R and TRKB co-precipitated, suggesting that these receptors may physically interact [19]. In fact, other studies have demonstrated the interaction between G-protein-coupled receptors and TRKB. For example, activation of cannabinoid receptor type-1 (CB1) transactivates TRKB [32,33] and increases TRKB-coupling to CB1 [32].

Our in vivo data indicates that C21 treatment counteracts the effects of shock-induced freezing response in wt mice but not in their BDNF.het littermates. The difference was only observed when animals were returned to the familiar context; freezing time did not change during the conditioning or in the unfamiliar context between the C21- and vehicle-treated groups. Consistent with our data, local administration of C21 into the central amygdala decreased fear response of animals exposed to either cued or contextual fear conditioning, an effect apparently mediated by AT2R-GABAergic neurons projecting to the periaqueductal gray [34]. Moreover, intrahippocampal administration of Ang II prevented the consolidation of aversive memory, an effect blocked by the AT2R antagonist PD123319 [35]. 

Since C21 treatment did not show any effect on BDNF.het mice, a BDNF-dependent mechanism may be plausible. Accordingly, it has been observed that BDNF.het animals do not respond to losartan administration in the forced swimming test; and in rats, the antidepressant-like effect of this compound is prevented by previous administration of k252a (a blocker of TRK receptors), suggesting again a BDNF/TRKB-dependent mechanism [19]. Therefore, our in vivo data further support such BDNF dependence of AT2R reducing the expression of conditioned fear. It is important to highlight that reduction in fear response is shared by many antidepressant drugs regardless of their primary mechanism of action. For instance, fluoxetine [17,18] or ketamine administration [18] reduced the freezing response in conditioned mice, and these effects were lost in BDNF.het mice [17] or in animals carrying a mutation in TRKB receptors that compromise drug binding [18].

Additionally, C21 was ineffective in the elevated plus-maze. Indeed, no effect of Ang II was observed on general locomotion or on anxiety-like behaviors in rats [35]. Corroborating our data, C21 was also ineffective on the elevated plus-maze when directly infused into the central amygdala, even though the same treatment protocol was sufficient to buffer the animal stress response [34]. 

Taken together, these outcomes raise the question whether the observed effects also apply to other drugs acting on RAS. The AT1R antagonist losartan was able to relieve ovariectomy-induced increase in anxiety-like behavior in the elevated plus-maze, and cognitive impairment in the novel object recognition test, and decreased the levels of plasma corticosterone [36]. Telmisartan, another AT1R antagonist, reduced the consequences of stress on cognitive impairment, and inhibited the activity of the HPA axis [37]. Telmisartan treatment also increased the level of BDNF in stressed animals and reduced anxiety-like behavior in the elevated plus-maze [38]. In contrast, losartan can decrease anxiety-like behavior in normo- and hypertensive animals, whereas the ACEi enalapril decreased anxiety-like behavior in hypertensive animals only [39]. Both AT1R antagonists and ACE inhibitors reduced the effects of scopolamine-induced amnesia in rats submitted to the elevated plus-maze to test for working memory [40]. Additionally, candesartan can relieve the stress response of rats when administered before the stressful event and decreases the level of glucocorticoids and other stress-related molecules with anxiolytic-like effect on non-stressed rats that submitted to the elevated plus-maze [41]. Therefore, there is a mix of data describing either an anxiolytic-like or null effect of drugs acting on RAS. However, the C21 lack of effect on the elevated plus-maze highlights the small influence, if any, of AT2R-acting drugs on basal anxiety levels.

## 4. Methods

### 4.1. Animals

Female adult mice (16–18 weeks old at the beginning of the experiments) of C57BL/6J-000664 background (from Jackson Laboratories, Bar Harbor, ME, USA, maintained in the Laboratory Animal Center of the University of Helsinki), carrying a deletion in one of the copies of Bdnf gene or wild-type littermates were used [17]. The animals were group housed (4–5/cage; type-II individually ventilated cages GM500, 391 × 199 × 160 mm, floor area 501 cm^2^; Tecniplast, Buguggiate, Varese, Italy) in a 12-h light/12-h dark cycle (light on at 7:00 a.m.), with free access to food and water except during the experimental sessions. All protocols were approved by the ethics committee for animal experimentation of Southern Finland (ESAVI/38503/2019).

### 4.2. Drugs

Compound 21 {butyl[3-(4-((1H-imidazol-1-yl)methyl)phenyl)-5-isobutylthiophen-2-yl] sulfonylcarbamate} (C21 sodium salt, MW= 497.61, purity= 97.7%; kindly donated by Vicore Pharma, Gothenburg, Sweden) and BDNF (Peprotech, #450-02, Rockhill, NJ, USA) were used. For in vitro experiments, C21 was dissolved in DMSO, and BDNF was dissolved in PBS; for in vivo experiments C21 was dissolved in sterile saline. Isoflurane (Vetflurane^®^, Virbac, Nice, France) was used for anesthesia in nose-to-brain administration of C21 (see below). 

### 4.3. Primary Cultures of Cortical Cells

Cultures of cortical cells from E18 rat embryos were prepared as previously described in detail [42]. Briefly, suspended cortical cells were seeded in poly-L-lysine-coated 24- (Corning, Tewksbury, MA, USA) or 96-well plates (View Plate 96, PerkinElmer, Waltham, MA, USA) at 250,000 or 60,000 cells/well, respectively. The cells were maintained in Neurobasal medium, supplemented with B27 and left undisturbed, except for medium change (1/3 twice per week). 

### 4.4. Surface TRKB

Cortical neurons were cultured in 96-well plates (60,000 cells/well, 100 μL/well of medium, DIV8) and treated with C21 (0, 0.1, 1, 10 μM/15 min) and surface levels of TRKB was determined as previously described [23,43]. Briefly, after drug administration, the wells were washed three times with cold PBS and fixed with 4% PFA for 20 min at room temperature (RT) under agitation. The cells were washed again three times with PBS for 5 min at RT and blocked for 1 h at RT (5% nonfat dry milk, 5% Bovine Serum Albumin-BSA-in PBS). Primary antibody against the extracellular portion of TRKB (R&D; #AF1494; 1:500) was added and incubated overnight (ON) at 4 °C. Following wash with PBS, the cells were incubated with anti-goat IgG HRP-conjugated antibody (Invitrogen; #61-1620; 1:5000) for 1 h at RT. The cells were washed four times with PBS (10 min at RT for each wash) and then ECL (1:1) was added to detect the signal by the plate reader (Varioskan Flash, Thermo Scientific, Waltham, MA, USA). The signal from the samples, after blank subtraction, were normalized by the average of the control group (C21 = 0) and expressed as percentage from control. 

### 4.5. Phospho-TRKB Interaction ELISA

For assaying the phosphorylated TRKB (pTRKB), cortical neurons were cultured in 24-well plates (250,000 cells/well, 500 μL/well of medium, DIV8) and treated with C21 (10 μM/15 min) and challenged with BDNF (0.1 ng/mL/15 min) at a concentration that is beyond the BDNF Ki for TRKB [18] but ineffective in activating the receptor in our conditions [33]. An independent cohort of cultured cortical cells was incubated with C21 (10 μM) for 3 days and challenged with BDNF (0.1 ng/mL/15 min).

The cells were washed with ice-cold PBS and lysis buffer [20 mM Tris-HCl; 137 mM NaCl; 10% glycerol; 0.05 M NaF; 1% NP-40; 0.05 mM Na3 VO4], containing a cocktail of protease and phosphatase inhibitors (Sigma-Aldrich, #P2714 and #P0044, respectively) was added.

The signal of pTRKB was determined by ELISA [18,23,44]. On the first day, a white 96-well plate (OptiPlate 96 F HB, White, PerkinElmer) was coated with primary antibody against the extracellular portion of TRKB (R&D; #AF1494; 1:1000 in carbonate buffer, pH 9.8, Na_2_CO_3_ 57.4 mM, NaHCO_3_ 42.6 mM) ON at 4 °C. The plate was blocked with 2% BSA in PBS with 0.1% Tween (PBS-T) for 2 h at RT. Then, cell lysates were added to the plate and left ON at 4 °C. The plate was washed again with PBS-T, and biotinylated anti-phosphotyrosine secondary antibody (BioRad; #MCA2472B; 1:2000) was added followed by ON incubation at 4 °C. Following another washing step, the plate was incubated with HRP-conjugated streptavidin (Thermo Scientific; #21126; 1:5000) for 2 h at RT. The plate was washed with PBS-T and the chemiluminescent signal was detected by a plate reader (Varioskan Flash, Thermo Scientific) after addition of ECL (1:1). The signal from the samples, after blank subtraction, were normalized by the average of the control group (ctrl/ctrl) and expressed as percentage from control. 

### 4.6. Behavioral Analysis

We used 34 mice (14 haploinsufficient to BDNF, BDNF.het, and 20 wt littermates) in the fear conditioning experiment. All treatment and behavioral experiments were conducted between 9:00–14:00 h. Littermates of wt or BDNF.het mice were divided into two groups, where they were given either vehicle or C21 (0.3 mg/kg/day, 0.009 μg/day; wt n = 8, 12; het n = 6, 8) by intranasal route [45]. Briefly, the animals were lightly anesthetized with isoflurane (2% in a chamber for 4 min) and 20 μL of drug solution was administered using a micropipette to the nostrils. The treatments were delivered once a day for 3 days, alternating nostrils to avoid irritation or lesions. The conditioning procedure started 2 h after the last C21 administration. Following a 5 min habituation, the animals received 3 scrambled shocks (0.6 mA/2s, intervals 30 s–1 min) in context A (conditioning, transparent walls 23 × 23 × 35 cm with metal grid bottom), followed by 2 min without any shocks, in a total of 10-min session. Next day, the mice were introduced to an unfamiliar context B (black walls with the same dimensions, and black sleek bottom) in a 5 min session. On the third day, the animals returned to the familiar context A for 5 min [46]. The time spent in freezing (s) was determined by the software (TSE, Bad Homburg, Germany) in the last 2 min of the conditioning session and in the full 5 min in the subsequent sessions.

An independent cohort of wild-type mice (12 wt) was submitted to the elevated plus-maze. The animals were divided (n = 6/group) and received the same drug administration protocol as described above (3 intranasal administrations of PBS or 0.3 mg/kg of C21) under light anesthesia. Two hours after the last administration the animals were submitted to the acrylic-built elevated plus-maze composed of two open arms (30 × 5 cm with a 1 cm rim) and two enclosed arms (same dimensions with a 25 cm wall) connected by a central platform (5 × 5 cm) and elevated 40 cm above the floor. The percentage of time and entries in the open arms and the number of entries in the enclosed arms and the total distance traveled were determined by the software (Ethovision XT 13, Noldus, Wageningen, The Netherlands) in the 5 min session.

### 4.7. Statistical Analysis

The data from surface TRKB assay were analyzed using Kruskal-Wallis test, followed by Dunn’s post hoc test since homoscedasticity was not observed among the groups. The effect-size of the treatment was calculated by Hedges’ method [47]. The data from pTRKB ELISA were analyzed by two-way ANOVA with C21 and BDNF as factors followed by Fisher’s LSD. The data from elevated plus-maze were analyzed by the unpaired Student’s *t* test. The data from fear conditioning were analyzed by two-way ANOVA with repeated measures with treatment, genotype and trials as factors, and the total distance traveled in the first 2 min of the conditioning session as covariant; since an interaction was observed a two-way ANOVA was performed for each trial, followed by Fisher’s LSD post hoc tests. *p* < 0.05 was considered significant.

## 5. Conclusions

Although still modest, there is increasing interest about the brain effect of RAS-acting drugs. Indeed, AT1R-antagonists such as losartan have been shown to improve stimuli discrimination, which was followed by changes in amygdala activity, in healthy volunteers with high trait anxiety [48]. Therefore, it is of interest to study the repurpose of these well-tolerated drugs for brain disorders with the goal to expand knowledge about the RAS in general, as new compounds could be generated with AT2R as target. In this sense, it seems that C21 is a promising candidate and further investigation into its behavioral effects is warranted. In conclusion, we observed that C21 can counteract the consequences of stress in a ‘non-anxiolytic manner’, possibly via changing in TRKB trafficking, thus resulting in an increase of TRKB on the cell surface and facilitation of BDNF-dependent behavioral changes.

## Figures and Tables

**Figure 1 pharmaceuticals-14-00773-f001:**
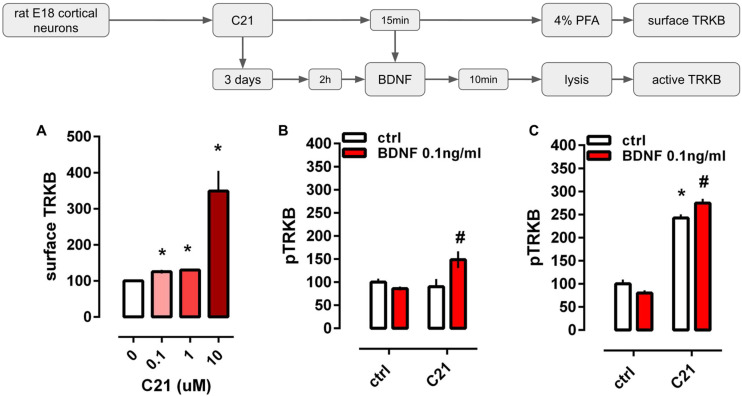
In vitro effects of C21 on TRKB surface exposure and activation. (**A**) Acute administration of C21 (shades of red bars, 0.1–10 μM/15 min) increased the surface exposure of TRKB in cultured cortical cells from rat embryos (*n* = 19–24/group). (**B**) Administration of C21 (10 μM/15 min) facilitated the effect of an ineffective dose of BDNF (0.1 ng/mL/10 min, red bars). (**C**) The longer administration of C21 (10 μM/3 days) increased pTRKB levels per se but still facilitated the effect of BDNF (0.1 ng/mL/10 min, red bars). Data expressed as mean ± SEM from ctrl/ctrl or 0 groups. * *p* < 0.05 from ctrl/ctrl or 0 group; ^#^ *p* < 0.05 from C21/ctrl (*n* = 8–12/group: Fisher’s LSD).

**Figure 2 pharmaceuticals-14-00773-f002:**
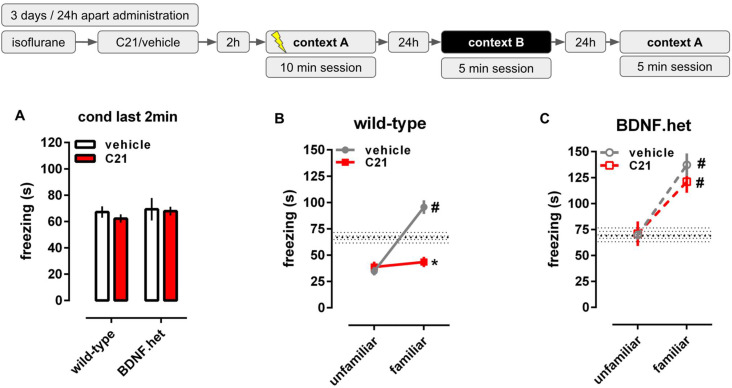
Effects of C21 in wild-type and BDNF haploinsufficient animals (BDNF.het) in contextual fear conditioning. (**A**) Administration of C21 (red bar, 0.3 mg/kg, intranasal) did not affect the freezing reaction at the end (last 2 min) of the conditioning session. (**B**) Treatment with C21 did not affect the freezing reaction in the unfamiliar context but decreased the freezing in familiar context on wild-type animals. Hatched area: mean ± SEM of the freezing time of vehicle-treated animals in the last 2 min of the conditioning session. (**C**) Treatment with C21 did not exert any effect in BDNF.het mice. Hatched area: mean ± SEM of the freezing time of vehicle-treated animals of each genotype in the last 2 min of the conditioning session. Data expressed as mean ± SEM of the time spent in freezing (s). * *p* < 0.05 from the vehicle-treated group at the same trial, ^#^ *p* < 0.05 from the freezing time in the unfamiliar context in the same group (*n* = 6–12/group).

**Table 1 pharmaceuticals-14-00773-t001:** Effect of C21 (0.3 mg/kg, intranasal) in the elevated plus-maze.

	Vehicle	C21
%OAT	45.08 ± 7.30	41.25 ± 7.22
%OAE	49.20 ± 5.39	46.01 ± 4.66
EAE (number)	54.50 ± 9.75	60.67 ± 5.47
dist trav (cm)	1838.28 ± 333.71	1833.49 ± 154.58

Data expressed as mean ± SD (*n* = 6/group).

## Data Availability

All data used in the present study were stored under a CC-BY license in FigShare (doi:10.6084/m9.figshare.12593396).

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
