# Peer review of "Facilitation of TRKB Activation by the Angiotensin II Receptor Type-2 (AT2R) Agonist C21"

_pharmaceuticals, 2021, doi:10.3390/ph14080773_

Round 1
Reviewer 1 Report
This ms. by Laukkanen et al. addresses a timely question regarding the effect of non-peptide AT2R agonist C21 on the activation of TRKB and conditioned fear. The results clearly show that C21 increased TRKB surface, but not pTKB levels in cell cultures. In vivo, it impaired conditioned fear, without affecting a standard measure of anxiety (i.e., behavior in the elevated plus maze).
The study is well performed and the results clear. I only have the following minor comments that can help improving what is already a very strong manuscript:
- Please, indicate the time of the day when the behavioral experiments took place expressed with regards to ZT.
- Particularly relevant with regards to the behavioral studies: please, indicate if they were performed in males and/or females, and precise age at the beginning of the experiments.
- The reporting and figure for the fear conditioning results is confusing. Why are the 2 last minutes analyzed? is freezing affected following shock exposure? If that is the case, what is the delta freezing between training/conditioning and testing? What is the difference between the familiar environment (fig 2B) and the conditioning environment? This part needs careful clarification in both main text and figure legend.
Author Response
Q1. Please, indicate the time of the day when the behavioral experiments took place expressed with regards to ZT.
R. We added the time of treatment and behavioral experiments in the Methods section: “All treatment and behavioral experiments were conducted between 9:00-14:00h.” Moreover, light/dark cycle was added to the Methods in the topic Animals - 12-h light/12-h dark cycle (light on at 7:00 am), thus making zeitgeber time readily implied.
Q2. Particularly relevant with regards to the behavioral studies: please, indicate if they were performed in males and/or females, and precise age at the beginning of the experiments.
R. All experiments were conducted in females. This was indicated in the methods section in the first versions of the manuscript. We edited to make it more clear: “Female adult mice (16-18weeks old at the beginning of the experiments) of C57BL/6J-000664 background (from Jackson Laboratories, maintained in the Laboratory Animal Center of the University of Helsinki), carrying a deletion in one of the copies of Bdnf gene or wild-type littermates were used”.
Q3. The reporting and figure for the fear conditioning results is confusing. Why are the 2 last minutes analyzed? Is freezing affected following shock exposure? If that is the case, what is the delta freezing between training/conditioning and testing? What is the difference between the familiar environment (fig 2B) and the conditioning environment? This part needs careful clarification in both the main text and figure legend.
R. The freezing was counted in the last 2min because it is the time where all the shocks were delivered, so we can have a better picture of the animal’s short-term response to the stressor. We changed the text to make it more clear: “Following a 5min habituation, the animals received 3 scrambled shocks (0.6mA/2s, intervals 30s-1min) in context A (conditioning, transparent walls 23×23×35cm with metal grid bottom), followed by 2min without any shocks, in a total of 10-min session.” Counting freezing during or between shock delivery is not a reliable approach in our experience, hence we prefer to use the described method as we did for similar protocols of fear conditioning, for instance see Karpova et al. 2011 [1]. Besides, since our protocol was designed to approach drug effect on the extinction trials, the most important measure in conditioning is the final freezing to show that animals ended the experiment with the same amount of conditioned response, in order to avoid any discrepancy in the fear salience that could likely remain until the next day and then confuse any relevant drug effect. Delta freezing between conditioning and testing was not described as we judge that fig2b speaks for itself inasmuch as the unfamiliar freezing (context unrelated to previous shocks) brings out the intrinsic basal freezing levels of the animals and the hatched area allows the reader to visually check for obvious differences in freezing between the first and second day. The main difference between the contexts are the color of the walls (transparent in familiar vs black in unfamiliar) and the grid floor (metal in familiar and black acrylic in unfamiliar). We have made it clear in the figure 2 workflow, highlighting context B (unfamiliar) as black to differentiate from the conditioning context. Additionally, both environments have been properly described within methods in the topic Behavioral Analysis.
References
1. Karpova, N.N.; Pickenhagen, A.; Lindholm, J.; Tiraboschi, E.; Kulesskaya, N.; Agústsdóttir, A.; Antila, H.; Popova, D.; Akamine, Y.; Bahi, A.; et al. Fear Erasure in Mice Requires Synergy between Antidepressant Drugs and Extinction Training. Science 2011, 334, 1731–1734.
Reviewer 2 Report
The review of the manuscript pharmaceuticals-1327049 entitled: Facilitation of TRKB activation by the angiotensin II receptor 2 type-2 (AT2R) agonist C21 by Laukkanen et al.
The paper deals with the effects of C21 on the activation of TRKB in vitro and conditioned fear in vivo. Authors reported the increased surface levels of TRKB and improved effect of BDNF on pTRKB after the administration of C21 in cultured cortical neurons from rat embryos. The effect of C21 on fear conditioning was noticed only in wt, but not in BDNF.het, mice and no effect of C21 on anxious behavior was noticed. The paper is very well written, and the reported results present important findings.
I only have some minor remarks.
line 16 - Please check the subscript in AT2R
line 172 – Please check the number of significant digits when presenting the results. While g for all the doses have 3 significant digits, there are four of them for 1uM
Figures are a bit confusing due to the presentation including the histogram and SEM together with squares and dots which, I assume, present individual values. Maybe authors should consider to choose between the two types of presentation (histograms in different colors or patterns with SEM or individual values), or at least, explain under the figures how the data are presented.
Author Response
Q1. line 16 - Please check the subscript in AT2R
R. Corrected.
Q2. line 172 – Please check the number of significant digits when presenting the results. While g for all the doses have 3 significant digits, there are four of them for 1uM
R. Corrected.
Q3. Figures are a bit confusing due to the presentation including the histogram and SEM together with squares and dots which, I assume, present individual values. Maybe authors should consider choosing between the two types of presentation (histograms in different colors or patterns with SEM or individual values), or at least, explain under the figures how the data are presented.
R. All data used in the present manuscript is available, thus we opted to represent the plots as bars only, as it makes the drug effect more clear. We thank the reviewer for this suggestion.
Reviewer 3 Report
Excellent paper, a pleasure to read. The only observation reggards the drug you used, C21. You should provide the chemical name.
Author Response
Q1. Excellent paper, a pleasure to read. The only observation regards the drug you used, C21. You should provide the chemical name.
R. We thank the reviewer for this compliment. We added the chemical name {butyl[3-(4-((1H-imidazol-1-yl)methyl)phenyl)-5-isobutylthiophen-2-yl] sulfonylcarbamate} in the methods section.